# Study of the TIG Welding Process of Thin-Walled Components Made of 17-4 PH Steel in the Aspect of Weld Distortion Distribution

**DOI:** 10.3390/ma16134854

**Published:** 2023-07-06

**Authors:** Marek Mróz, Bartłomiej Kucel, Patryk Rąb, Sylwia Olszewska

**Affiliations:** 1Department of Foundry and Welding, Faculty of Mechanical Engineering and Aeronautics, Rzeszow University of Technology, Al. Powstańców Warszawy 12, 35-959 Rzeszow, Poland; p.rab@prz.edu.pl (P.R.); s.olszewska@prz.edu.pl (S.O.); 2MB Aerospace Rzeszow, ul. Przemysłowa 9b, 35-105 Rzeszow, Poland; bartlomiej.kucel@mbaerospace.com

**Keywords:** TIG method, 17-4 PH steel, thin-walled components, weld distortion

## Abstract

This article presents the results of a study on the distribution of weld distortion in thin-walled components made of 17-4 PH steel, resulting from TIG (Tungsten Inert Gas) welding. Both manual and automatic welding processes were examined. Physical simulation of the automated welding process was conducted on a custom-built welding fixture. Analysis of weld distortion in thin-walled components made of 17-4 PH steel was based on the results of measurements of transverse shrinkage and displacement angle values. These measurements were taken on thin-walled parts before and after the welding process using a coordinate measuring machine (CMM). To determine the effect of manual and automated welding processes on the microstructure of the welded joint area, metallographic tests and hardness measurements were performed. The microstructure was analyzed using a scanning electron microscope (SEM). An analysis of the chemical composition of selected welded joint zones was also conducted. These tests were performed using an optical emission spectrometer (OES). According to the results, the use of automated welding and special fixtures for manufacturing thin-walled aircraft engine components made of 17-4 PH steel reduces the propensity of these components for distortion due to the effects of the thermal cycle of the welding process. This conclusion is supported by the results of the observation of the microstructure and analysis of the chemical composition of the various zones of the welded joint area.

## 1. Introduction

The aerospace industry, especially in the design of aircraft engines, frequently employs welded structures made of sheet metal. These designs are distinguished by their lower overall weight, which improves the engine’s thrust-to-weight ratio [1,2]. During the manufacturing process of welded structures, the material is exposed to the effects of the welding thermal cycle. This exposure can result in irreversible changes in shape and dimensions due to welding stresses and strains [3,4,5]. The intensity of these changes also depends on the effect of temperature on the values of such material parameters as longitudinal modulus of elasticity, transverse modulus of elasticity, Poisson’s ratio, yield strength, and coefficient of linear thermal expansion. As temperature increases, the values of the longitudinal modulus of elasticity and yield strength decrease. Research on materials that exhibit high strength at minimum yield strengths is reported in papers [6,7,8].

In the welding process, rapid heating and rapid cooling occur in the thermal cycle of the welded joint area. This causes a mechanical interaction between the different areas of the welded joint. This process is further intensified by the occurring phase transformations (solid state transformations in the heat affected zone). In specific areas of the welded joint, changes in the dimensions and shape of structural elements may occur, resulting from the expansion or contraction of the welded material and the weld. These strains can be longitudinal, transverse, or angular. Longitudinal strains are caused by the shrinkage of the weld and the adjacent material in the direction of the length of the welded joint [9]. Angular strains are present in all types of joints and welds. They are induced by the non-uniform transverse shrinkage of the weld in its thickness and shrinkage of the metal adjacent to the weld. The measure of an angular strain is the β angle, which is the angle of refraction of the joint plane. The method of determining this angle using numerical methods and neural networks is described in paper [10], among others.

Numerical simulation is one of the modern tools for predicting welding stresses and distortion. The technical literature reports studies on the use of various programs, such as SYSWELD [11,12,13,14,15], ANSYS [16,17,18], ABAQUS [19,20], or AUTODESK CFD [21], for modelling welding stress and distortion distributions.

A significant factor affecting the formation of weld distortion is the type of heat source used in the welding thermal cycle. Proper selection of process parameters, especially the current parameters of the heat source, can help minimize weld strains. This is demonstrated in papers on GTAW welding [22,23] and GMAW welding [11,24], among other works. Similar conclusions were also reached by the authors of paper [25] on the FSW method, and papers [5,26,27] dealing with laser welding.

One of the ways to minimize weld distortion is to establish a proper welding procedure that accounts for the complexity of the welded structure, as well as the sequence and manner of making the welds. Paper [11] presents the results of a numerical simulation of different variants of back-step welds with respect to minimizing weld distortion.

The challenges of achieving adequate dimensional tolerance and shape tolerance of a welded structure are particularly evident in the case of thin-walled components, which are extensively utilized in the construction of aircraft engines (aerospace industry). As the aviation industry developed, new materials and design solutions were adopted. This led to changes in the manufacturing process to ensure sufficiently high performance properties of aviation parts [28]. 

Turbofan and turbojet engines can be divided into two sections, which differ in operating temperature: a cold section and a hot section [29,30]. Besides the required propulsion performance, the engines are characterized by compact design, low weight, reliable operation, and durability. The high demand for these designs translates into the complexity of engine components and the manufacturing processes for their production, as well as the ongoing search for new solutions and materials. The primary focus is on high strength and resistance to corrosive environments at elevated temperatures [31,32].

One of the materials employed in modern aircraft engines is low-carbon martensitic steel 17-4 PH (X5CrNiCuNb16-4 according to EN 10088-2 or UNS S17400 according to ASTM A564). This 17-4 PH steel, a registered trademark of AK Steel, is distinguished by high strength, excellent resistance to high-temperature corrosion, relatively high hardness, substantial resistance to thermal fatigue, and good durability of welded joints. Because of its properties, the steel has also been employed in industrial machinery, power, shipbuilding, geothermal, and petrochemical applications. In the aviation industry, it is used for elements of aircraft equipment and in the construction of aircraft engines for elements of the so-called engine cold zone. These include fan housings and compressor housings [33,34,35,36,37,38]. 

The principal alloying elements of 17-4 PH steel are chromium and copper. The steel is considered to have high corrosion resistance due to the presence of more than 12% chromium, while the 4% copper content improves corrosion resistance and increases heat resistance and creep resistance [39]. The authors of paper [40] emphasize the high corrosion resistance of 17-4 PH steel, arguing that, in terms of corrosion resistance in various environments, this steel is comparable to austenitic stainless steel 304 and group 400 stainless steel alloys. According to the authors of papers [40,41,42,43], this material is easily weldable by conventional arc welding methods such as TIG, laser welding, and hybrid methods. The authors of paper [44,45] state that the risk of hot cracks in the weld is low due to the solidification of the weld with the formation of δ-ferrite.

The authors of papers [25,46,47] have demonstrated that, by applying appropriate heat treatment, the hardness of 17-4 PH steel can be significantly increased. The treatment primarily involves supersaturation at high temperatures and aging at lower temperatures. The hardness of the 17-4 PH steel after treatment amounted to 49 HRC.

The research in this article focuses on the analysis of the effect of the type of welding process flow (manual or automated welding) on the propensity for distortion that arises as a result of the impact of the thermal cycle of welding thin-walled components made of 17-4 PH steel. This study introduces an original fixture solution for automatic TIG welding of these components. To fully illustrate the impact of the thermal cycle of manual and automatic TIG welding of thin-walled 17-4 PH steel elements, microstructure tests and hardness measurements were conducted.

## 2. Materials and Methods

The test material used was 0.9 mm thick 17-4 PH steel sheet. The chemical composition of the steel, as determined using a Bruker Q4 TASMAN (Kalkar, Germany) emission spectrometer, is shown in Table 1 (average of 5 analyses). The chemical composition of steel 17-4 PH (1.4021) according to EN 10088-2 is as follows (%wt.): max 0.07 %C, max 0.07 %Si, max 1.50 %Mn, max 0.04 %P, max 0.015 %S, 15.00–17.00 %Cr, 3.00–5.00 %Cu, max 0.06 %Mo, 3.00–5.00 %Ni, 5x%C %Nb, remainder Fe.

Strips 50 mm wide and 150 mm long were cut from the sheet metal, rolled up, and joined by a butt weld to produce a pipe section with a diameter of 150 mm. The test joint comprised two pipe components connected by a circumferential weld (one bead). Prior to making the test joint, the components were cleaned and degreased.

The test joints were made using the TIG method in two variants: manual and automated. Table 2 summarizes the basic parameters of TIG welding. 

The process of automated welding of the test joint was implemented on the bench shown in Figure 1, which includes a special welding fixture with a welding gun as its main component. This gun, equipped with copper cooling pads, also ensured the protection of the test joint root by flushing it with argon. The gun with the tacked tubular elements was mounted on a swivel. A TIG welding torch with automated wire feeding (additional material) was used to create the joint. The torch utilizes a 2% cerium tungsten electrode (WC 20) with a diameter of 3.2 mm and a sharpening angle of 30°.

Before starting the welding process, the tacked test joints were measured using a Mitutoyo Crysta 122010 CMM (Tokyo, Japan) with a Renishaw REVO-2 measuring head (Gloucestershire, UK). Figure 2 depicts a diagram of the distribution of measurement points in lines A, B and C. These measurements served as a reference for estimating the values of transverse shrinkage, angular strain, and profile deformation.

The methodology for determining angular strain values is shown in Figure 3. The strain angle is the angle of intersection of the trend line of the transverse profile based on points in lines A, B and C. The weld areas and sample edges were ignored in the calculations. Transverse shrinkage for both welding variants was also measured.

Visual examinations were performed based on PN-EN ISO 5817 to identify potential inconsistencies in the test welded joints. 

Metallographic tests were conducted to evaluate the changes in the microstructure of the joined components, with a particular focus on the impact of the welding thermal cycle on the structure and size of the heat affected zone (HAZ). Microstructure studies of test welded joints were performed on metallographic specimens. The metallographic specimens were prepared by cutting from the joints on a Struers Labotom-3 metallographic cutter (Copenhagen, Denmark), then ground and polished with abrasive paper, polishing cloths, and diamond slurries on a Metimex SM-PM 250AV1 laboratory grinder-polisher (Pyskowice, Poland). The specimens were etched with Kalling’s reagent for 5 s for microstructure revelation, followed by observation under a Tescan VEGA 3 scanning microscope (Brno, Czech Republic). 

Hardness measurements were also made on the test joints, in accordance with the scheme shown in Figure 4.

Hardness measurements were made using the Vickers method, with the HV5 load. These measurements were carried out using a Zwick-Roell ZHV10 hardness tester (Ulm, Germany).

## 3. Results and Discussion

### 3.1. Measurements of Transverse Strain and Angular Strain

Plots of transverse profiles, which are an illustration of the transverse strain at the measurement points in lines A, B and C on the test joint made manually and the difference between the transverse profiles before and after welding, are shown in Figure 5, Figure 6 and Figure 7.

Plots of transverse profiles at the measurement points in lines A, B and C on the test joint made automatically and the difference between the transverse profiles before and after welding are shown in Figure 8, Figure 9 and Figure 10.

To better demonstrate the impact of the welding method (manually or automatically), Figure 11 and Figure 12 present the transverse profiles determined on lines A, B and C on manually and automatically welded test joints. These waveforms indicate clear differences in the distribution of transverse shrinkage values of test joints welded manually and automatically.

In the case of the manually made test joint, the connected sheets across their width are noticeably more corrugated. The wave-like character of the distribution of transverse shrinkage is particularly evident at the beginning and end of the transverse profile on lines A and C. In the case of line C, this profile has a rising character from the edges of the joined components in the direction of the weld, reaching peaks of distortion on both sides of the weld. This testifies to the greater distortion of test joints caused by the thermal cycle of the manual welding process, compared to automated welding.

The transverse profiles of the automatically made test joint show a similar pattern on both sides of the weld, with an increasing tendency of the transverse shrinkage value in the direction from the edges of the components to be joined to the weld. For this joint, there is no distinct corrugation of the sheet metal.

The obtained results indicate that the thermal cycle of automated welding has a significantly decreased effect on the distortion of the welded joint, compared to a joint made by hand. The outcome of the automation of the welding process was a correction of the process parameters of the TIG method, resulting in a reduction in the linear energy of the welding process (Table 2). This was reflected in the reduced amount of heat introduced into the welded joint area, resulting in a lower impact of the welding thermal cycle.

The results of measuring the angular strain values of test joints made manually and automatically are given in Table 3.

The average value of the angular strain *β* of the test joint made manually is 0.12 °C, while that for the test joint made by automated welding is 0.11 °C. Once again, the automatically welded joint has a lower propensity for angular strain *β*, compared to the joint made by hand.

### 3.2. Visual Testing 

The purpose of visual testing (VT) was to evaluate the welded joints for the presence of weld discontinuities. An example view of the weld face of the test joints is presented in Figure 13. In both cases, observing the weld face and root did not reveal any significant welding inconsistencies. However, it was noted that the face of the weld made by the automated method has better smoothness and evenness and the weld is continuous. In the case of a weld made by hand, the way the weld is made with a back-step technique (sectional welding) and small deviations in the straightness of the weld are visible.

### 3.3. Macro- and Microstructure Testing

The macrostructure of the welded joint area made manually is shown in Figure 14. 

The microstructure of the weld, the heat affected zone, and the parent material were analyzed in this joint. The area of the heat affected zone was analyzed on the basis of the extent of the impact of the welding thermal cycle. Consequently, four areas (A–D) were revealed in the heat affected zone. 

The sample microstructure of the weld, the A–D areas of the heat affected zone and the parent material of the test joint made by hand are shown in Figure 15, Figure 16, Figure 17, Figure 18, Figure 19 and Figure 20.

The macrostructure of the welded joint area made by the automated welding method is shown in Figure 21. Analysis of the macrostructure of the area of this joint suggests a similar structure, compared to the joint made manually. The heat affected zone of the joint also displayed areas of varying microstructure, resulting from the extent of the welding thermal cycle. Thus, as in the case of the manually welded joint, four areas (A, B, C and D) were identified in the heat affected zone of the test joint made by the automated welding method.

The sample microstructure of the weld of areas A–D of the heat affected zone of the test joint made by the automated welding method is shown in Figure 22, Figure 23, Figure 24, Figure 25, Figure 26 and Figure 27.

The microstructure of the parent material (Figure 22 and Figure 27), i.e., 17-4 PH steel, is martensitic, with a small number of very fine, almost invisible carbides located at grain boundaries. 

Analysis of the test welded joints made manually and automatically showed a great similarity in the effect of the welding thermal cycle on the microstructure of their respective areas. 

The weld, in both joints, reveals precipitates of ferrite δ and martensite. However, the weld of the manually welded joint showed a large number of fine carbides. It can be assumed that the thermal cycle of welding is responsible for the carbide release process. The manually welded joint was made with a back-step technique, i.e., sectional weld. After making the first section, the thermal cycle of the next section caused heating of the previously made weld which resulted in the process of carbide release in the weld from the supersaturated solution of carbon in iron α. In this way, laying each successive weld section intensified the secretion processes in the previously laid sections. 

Upon examining the microstructure of the heat affected zone (HAZ), ferrite δ and martensite are immediately adjacent to the fusion line (area A). Areas B and C exhibit the typical structure of quenching products (martensite and tempered martensite). As we move away from the weld toward the parent material, there are small amounts of tempered martensite in the D area. This is attributable to the lower intensity of the impact of the thermal cycle of the welding process. 

Carbides were observed in the heat affected zone of the manually welded joint (as in the weld) (Figure 22, Figure 23, Figure 24 and Figure 25). The intensity of the carbide release process in individual areas A, B, C and D varied depending on the intensity of the impact of the thermal cycle of the welding process. The highest amount of carbides was witnessed in areas closer to the weld.

The results of X-ray microanalysis of the chemical composition of the carbides are presented in Table 4.

From the obtained results, it was found that the identified carbide phases mainly contain chromium, nickel and copper. The iron content results from carbides that are too small and electron beam scanning of areas next to these carbides. Even though the determination of carbon content by X-ray microanalysis is not accurate (carbon contamination phenomenon), Table 4 provides the carbon content to confirm that the phases analyzed are carbides.

The analysis of the welded joint made by the automated welding method shows a similar effect of the thermal cycle of the welding process on the microstructure of its individual areas, with the exception that carbides were not found in such large numbers in both the weld and the heat affected zone. This indicates the smaller extent and lower intensity of the impact of the thermal cycle of the welding process. This is further supported by measurements of the width of the entire heat affected zone and its individual areas, as shown in Table 5. The total width of the heat affected zone for the joint made by automated welding is almost half that of the test joint made by hand.

### 3.4. Hardness Measurements

The hardness measurements results of test welded joints are presented in Table 6 and in Figure 28.

The obtained results indicate differences in hardness values in individual areas of welded joints made manually and automatically. In the case of an automatically produced joint, due to the less intensive impact of the heat cycle of the welding process, the hardness distribution in individual areas of the heat affected zone, compared to the weld and the base material, is more stable than in a manually made joint. However, in both joints, area C of the heat affected zone witnessed a large decrease in hardness, dropping to about 340 HV5. This indicates that the most significant microstructural changes occurred in area C due to the thermal cycle of the welding process. A more detailed explanation of these changes, given the thematic scope of this publication, will be the subject of a subsequent publication. The publication will employ the results of additional research, including color metallography results, to reveal the percentage of martensite or other hardening products in individual HAZ areas, which form due to the thermal cycle of the welding process.

## 4. Conclusions

Not only does the type of welding method used exert a significant impact on the size and distribution of welding deformations, but so does the method of implementation. Replacing manual welding of thin-walled 17-4 PH steel elements with automatic welding led to a reduction in the amount of heat introduced into the joined elements (lower linear energy of the welding process). Consequently, the deformation of the automatically welded joint was less than that of the manually welded joint. In the case of a manually welded joint, significant ripples appeared in the areas extending from the weld to the parent material.

The positive effect of automatic welding on weld quality, especially on the condition of the weld face and the weld root surface, was confirmed.

The microstructure test results of welded joints corroborate the lower intensity of the heat cycle of the welding process in an automatically welded joint. In this joint, a heat affected zone of a similar structure was found as in a manually welded joint, but with a significantly smaller HAZ width. Both welded joints contain areas with a similar microstructure, featuring quenching and tempering products resulting from heating in the thermal cycle of the welding process and subsequent rapid cooling. Furthermore, the presence of carbides was observed in the microstructure of the manually welded joint.

The welding method profoundly influences the hardness distribution in individual areas of welded joints. In automatic welding, the HV5 hardness distribution is more stable compared to the hardness distribution in a manually welded joint, reaffirmed by the lower intensity of the thermal cycle of the welding process on the microstructure of the heat affected zone in the automatically welded joint. The hardness values in the HAZ of an automatically welded joint are closer to the hardness of the weld and the parent material, compared to the HAZ of a manually welded joint. It can be deduced that, in the case of an automatically welded joint, the intensity of various types of hardening and tempering processes caused by the thermal cycle of the welding process is lower compared to a manually welded joint.

Automated TIG welding is recommended for manufacturing 17-4 PH thin-walled components. This is particularly critical when such components are used in the construction of aircraft engines.

## Figures and Tables

**Figure 1 materials-16-04854-f001:**
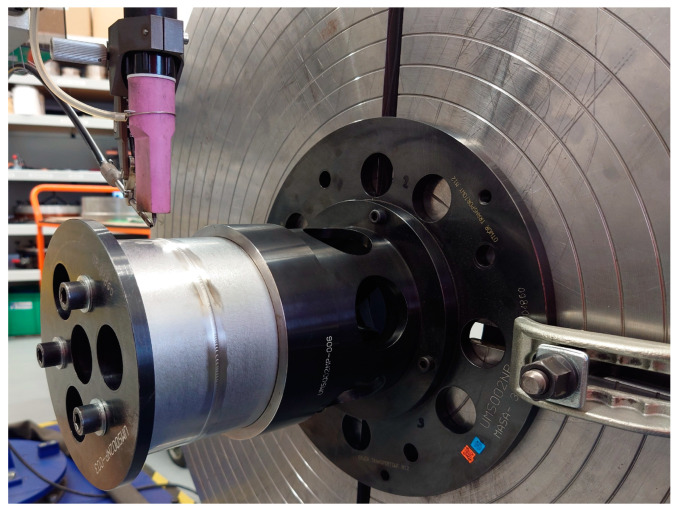
Bench for automated TIG welding of thin-walled components made of 17-4 PH steel.

**Figure 2 materials-16-04854-f002:**
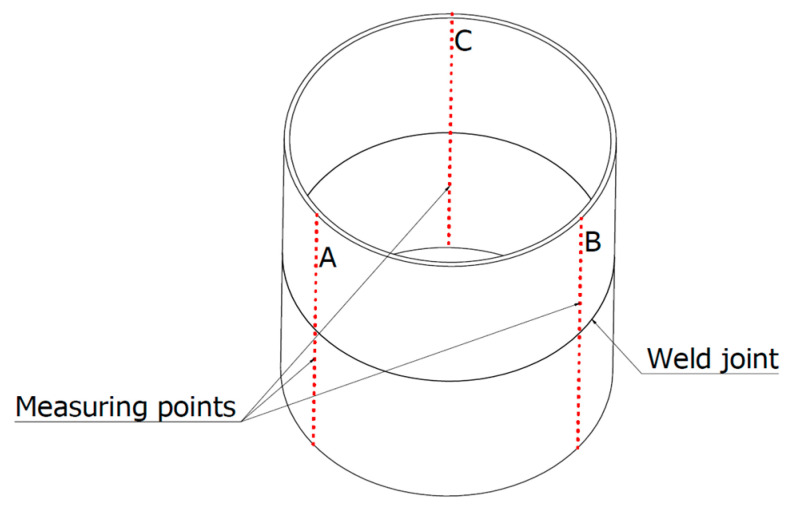
Diagram of the distribution of measurement points.

**Figure 3 materials-16-04854-f003:**
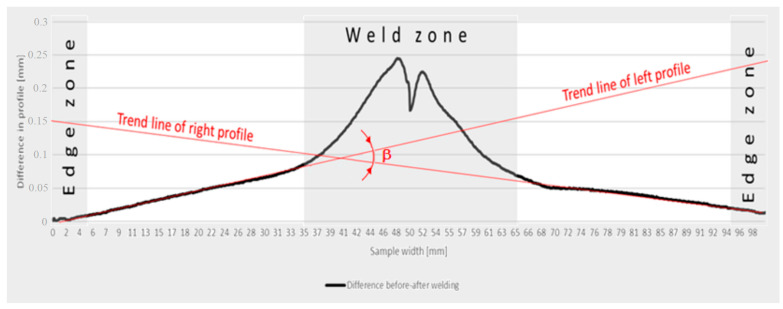
Methodology for calculating angular strain values.

**Figure 4 materials-16-04854-f004:**
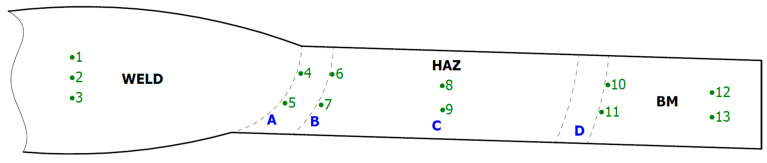
Scheme of hardness measurements of test welded joints manually and automatically.

**Figure 5 materials-16-04854-f005:**
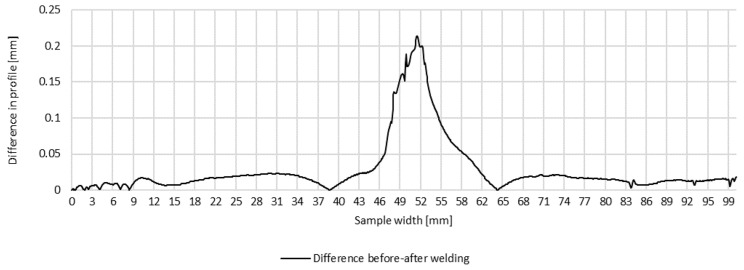
Difference in transverse profiles on line A before and after manual welding.

**Figure 6 materials-16-04854-f006:**
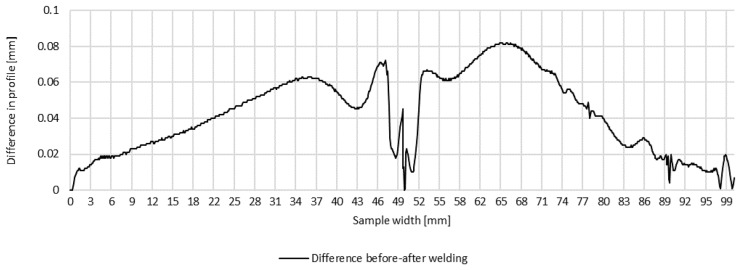
Difference in transverse profiles on line B before and after manual welding.

**Figure 7 materials-16-04854-f007:**
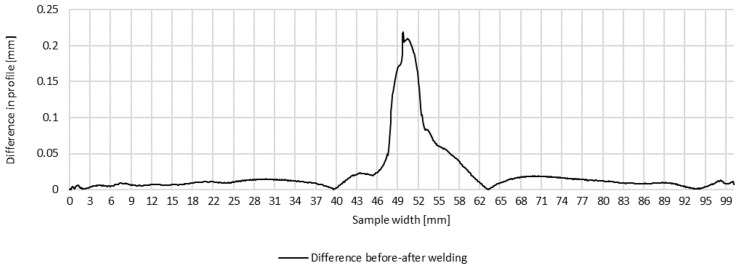
Difference in transverse profiles on line C before and after manual welding.

**Figure 8 materials-16-04854-f008:**
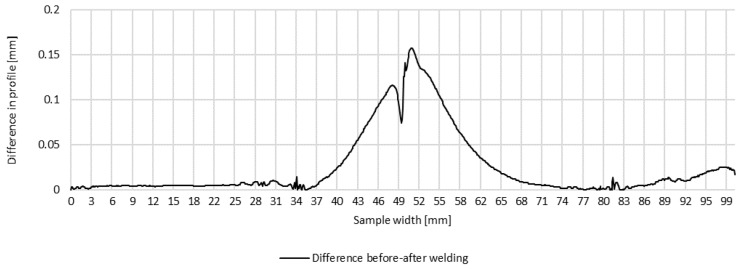
Difference in transverse profiles on line A before and after automated welding.

**Figure 9 materials-16-04854-f009:**
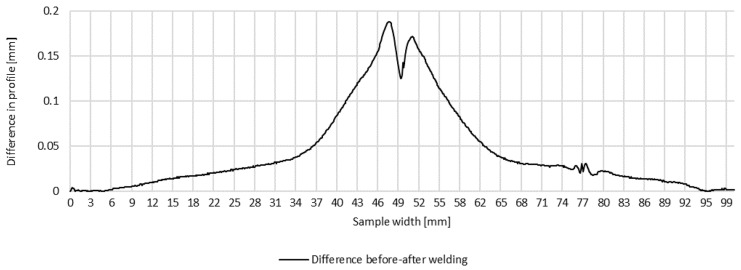
Difference in transverse profiles on line B before and after automated welding.

**Figure 10 materials-16-04854-f010:**
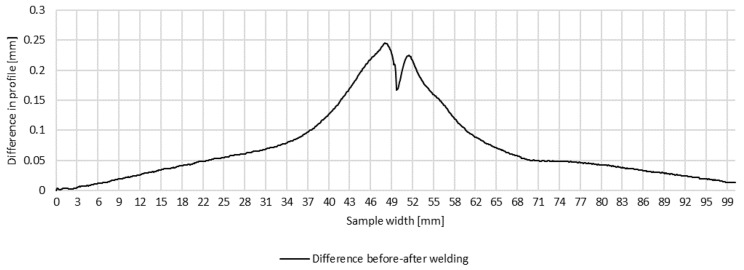
Difference in transverse profiles on line C before and after automated welding.

**Figure 11 materials-16-04854-f011:**
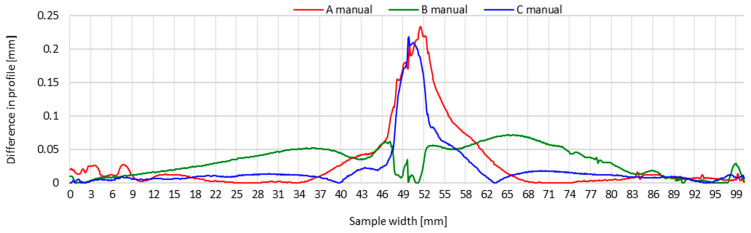
Transverse profiles on lines A, B and C after making a test joint of thin-walled components of 17-4 PH steel—manual welding.

**Figure 12 materials-16-04854-f012:**
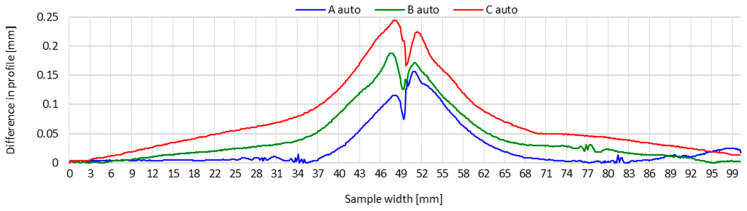
Transverse profiles on lines A, B and C after making a test joint of thin-walled components of 17-4 PH steel—automated welding.

**Figure 13 materials-16-04854-f013:**
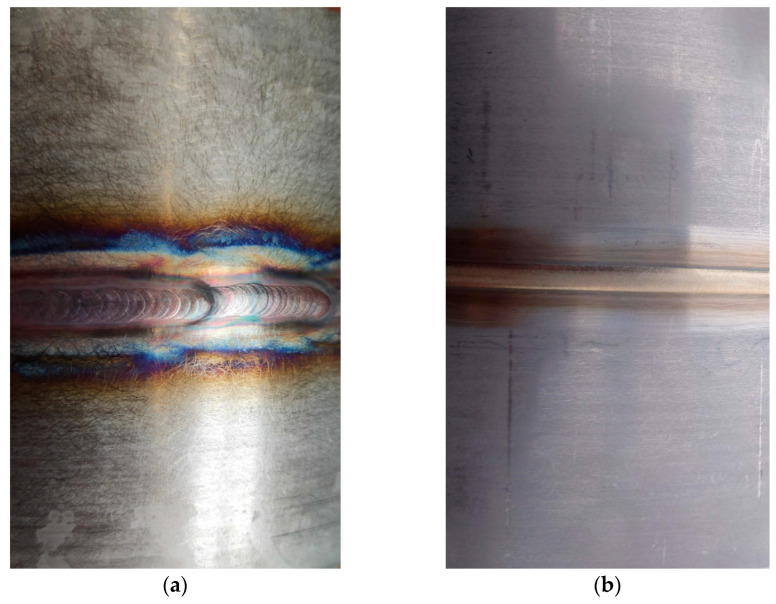
Example view of the face of the test joint made manually (**a**) and by automated welding method (**b**).

**Figure 14 materials-16-04854-f014:**
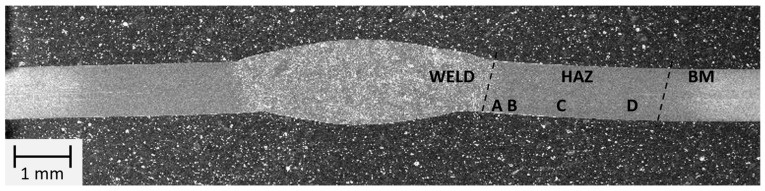
Macrostructure of the welded joint area of thin-walled 17-4 PH steel components made manually by TIG method.

**Figure 15 materials-16-04854-f015:**
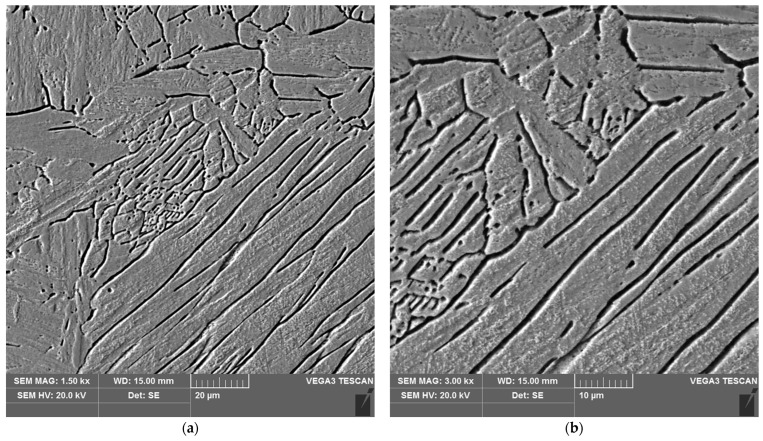
Microstructure of the weld area of the test joint made manually: (**a**) magnification 1500×, (**b**) 3000×.

**Figure 16 materials-16-04854-f016:**
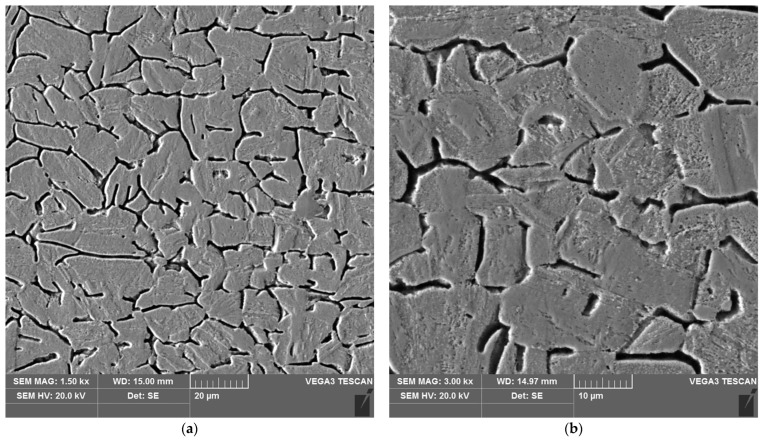
Microstructure of the heat affected zone in area A of the test joint made manually: (**a**) magnification 1500×, (**b**) 3000×.

**Figure 17 materials-16-04854-f017:**
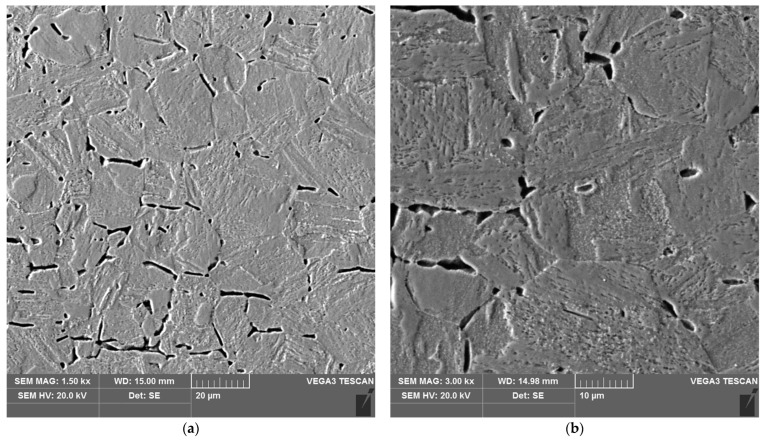
Microstructure of the heat affected zone in area B of the test joint made manually: (**a**) magnification 1500×, (**b**) 3000×.

**Figure 18 materials-16-04854-f018:**
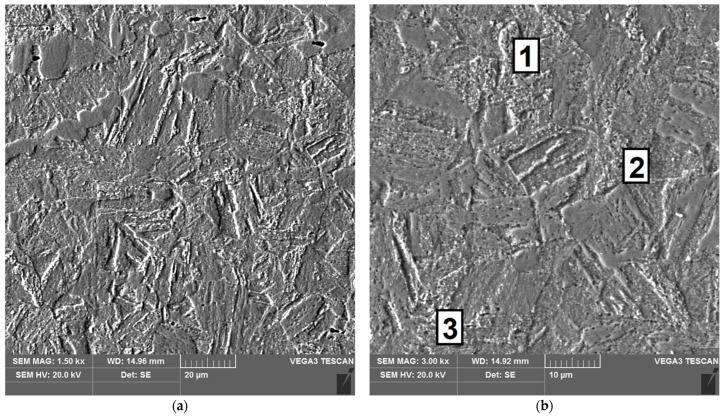
Microstructure of the heat affected zone in area C of the test joint made manually: (**a**) magnification 1500×, (**b**) 3000×. 1–3 are the places where the analysis of the chemical composition was performed.

**Figure 19 materials-16-04854-f019:**
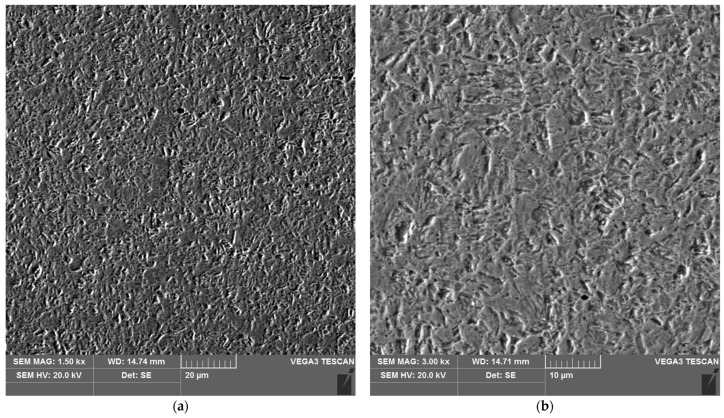
Microstructure of the heat affected zone in area D of the test joint made manually: (**a**) magnification 1500×, (**b**) 3000×.

**Figure 20 materials-16-04854-f020:**
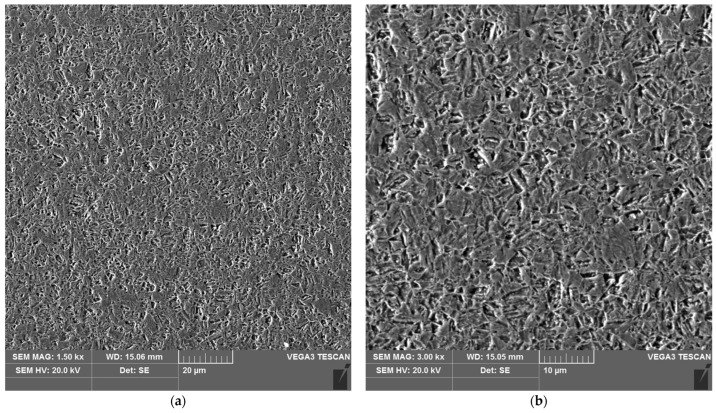
Microstructure of the parent material area of the test joint made manually: (**a**) magnification 1500×, (**b**) 3000×.

**Figure 21 materials-16-04854-f021:**
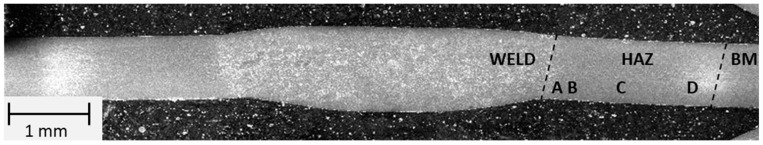
Macrostructure of the welded joint area of thin-walled 17-4 PH steel components made automatically by TIG method.

**Figure 22 materials-16-04854-f022:**
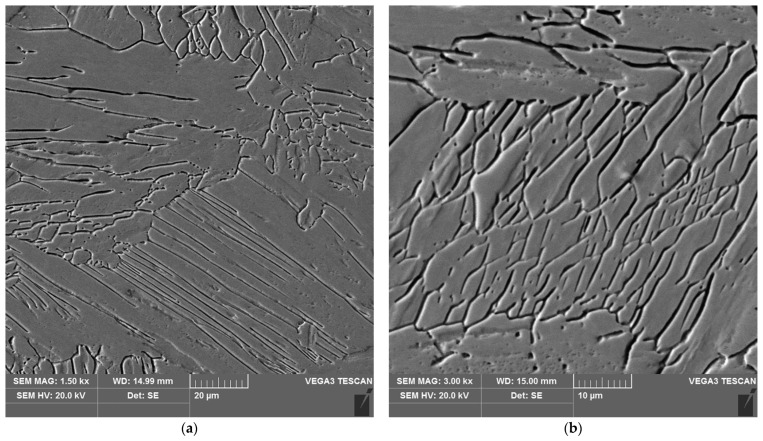
Microstructure of the weld area of the test joint made by the automated welding method: (**a**) magnification 1500×, (**b**) 3000×.

**Figure 23 materials-16-04854-f023:**
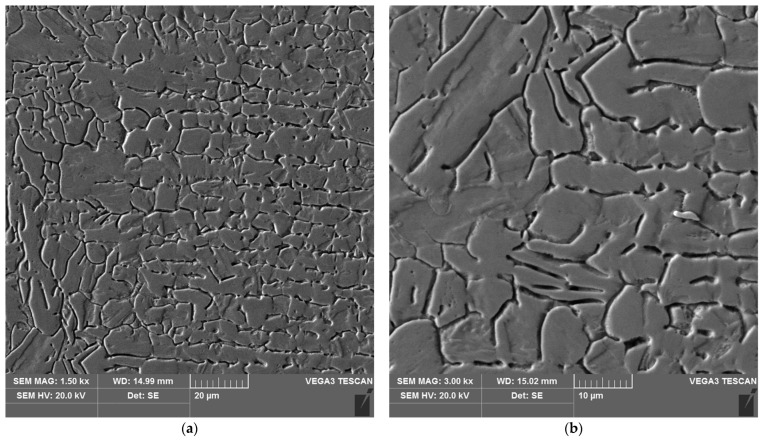
Microstructure of the heat affected zone in area A of the test joint made by the automated welding method: (**a**) magnification 1500×, (**b**) 3000×.

**Figure 24 materials-16-04854-f024:**
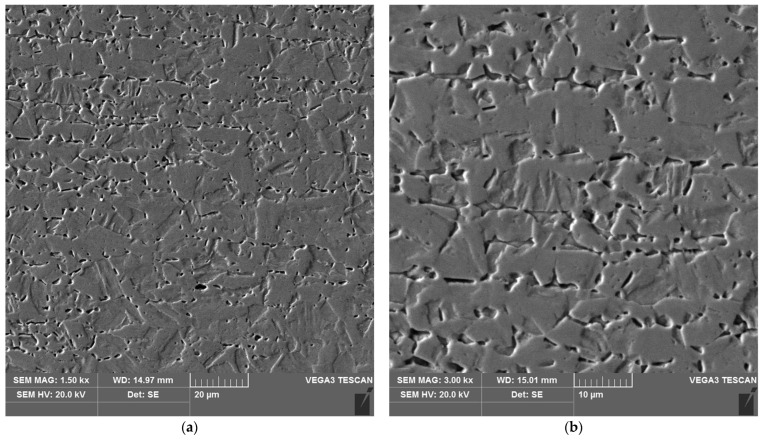
Microstructure of the heat affected zone in area B of the test joint made by the automated welding method: (**a**) magnification 1500×, (**b**) 3000×.

**Figure 25 materials-16-04854-f025:**
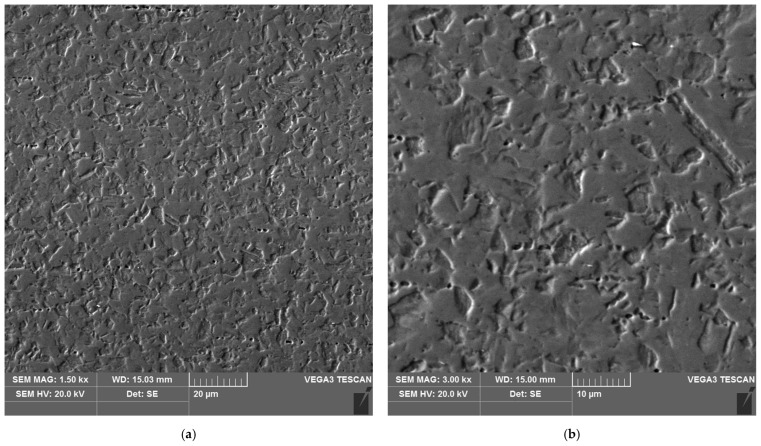
Microstructure of the heat affected zone in area C of the test joint made by the automated welding method: (**a**) magnification 1500×, (**b**) 3000×.

**Figure 26 materials-16-04854-f026:**
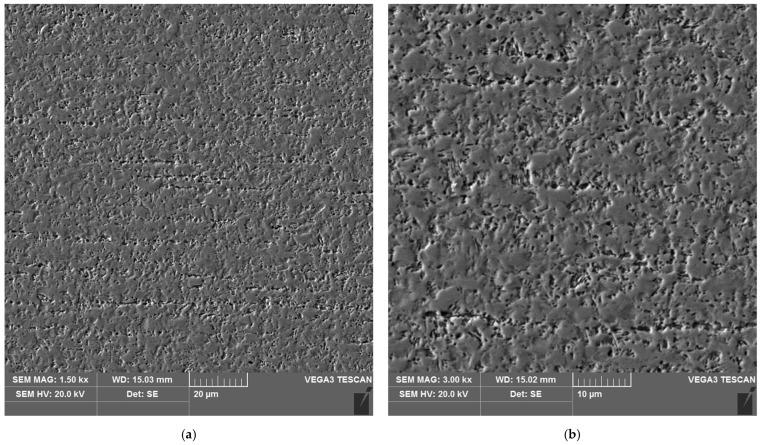
Microstructure of the heat affected zone in area D of the test joint made by the automated welding method: (**a**) magnification 1500×, (**b**) 3000×.

**Figure 27 materials-16-04854-f027:**
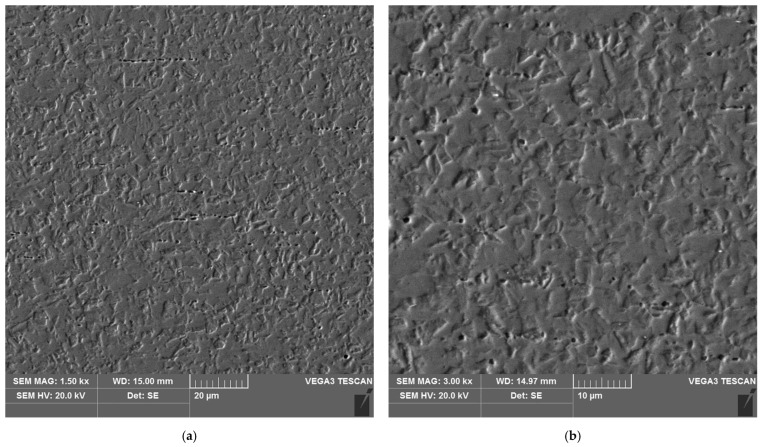
Microstructure of the parent material area of the test joint made by the automated welding method: (**a**) magnification 1500×, (**b**) 3000×.

**Figure 28 materials-16-04854-f028:**
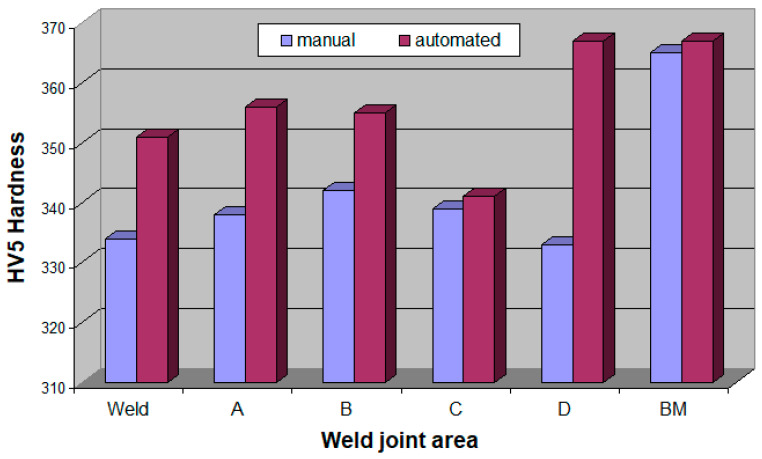
Hardness of the test joint made by the automated welding method.

**Table 1 materials-16-04854-t001:** Chemical composition of 17-4 PH steel.

Chemical Compositions (%wt.)							
C	Si	Mn	P	S	Cr	Mo	Ni
0.052	0.376	0.511	0.084	0.0081	16.18	0.075	3.857
Cu	Al	Co	Nb	Ti	V	W	Fe
4.092	0.0068	0.048	0.329	0.0075	0.059	0.029	74.24

**Table 2 materials-16-04854-t002:** Process parameters of TIG welding of thin-walled components of 17-4 PH steel.

Parameter	Automated	Manual
Amperage I (A)	34	37
Voltage U (V)	~8.3	~8.1
Welding speed v (mm/min)	170	65
Linear energy (kJ/cm)	1.00	2.8

**Table 3 materials-16-04854-t003:** Angular strain values of test joints made manually and automatically.

Welding Method	Measurement Point	Angular Strain *β* (Degrees)
Manual	A	0.02
	B	0.13
	C	0.21
Automated	A	0.04
	B	0.26
	C	0.04

**Table 4 materials-16-04854-t004:** Results of chemical composition analysis of carbides (Figure 18b).

Point	Chemical Composition [%wt.]				
	C	Cr	Ni	Cu	Fe
1	2.87	16.69	4.30	2.76	73.39
2	2.17	16.64	4.54	3.10	73.55
3	1.73	16.83	4.53	2.94	73.97

**Table 5 materials-16-04854-t005:** Results of measurements of the width of individual areas of the heat affected zone of test joints made manually and by automated welding.

Welding Method	Measurement Point	Width of the Heat Affected Zone Area [μm]
manually	A	120–150
	B	90–105
	C	1800
	D	2000
automatically	A	90–100
	B	80–90
	C	900
	D	1100

**Table 6 materials-16-04854-t006:** HV5 hardness measurements results.

Welding Method	Hardness Measurement Point (HV5)												
	1	2	3	4	5	6	7	8	9	10	11	12	13
Manual	330	336	366	338	338	344	340	338	340	330	336	366	364
average value	334 in weld			338 in area A		342 in area B		339 in area C		333 in area D		365 in BM	
Automated	350	348	355	354	358	356	354	342	340	366	368	368	366
average value	351 in weld			356 in area A		355 in area B		341 in area C		367 in area D		367 in BM	

## Data Availability

Data is contained within the article.

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
