# Peer review of "Study of the TIG Welding Process of Thin-Walled Components Made of 17-4 PH Steel in the Aspect of Weld Distortion Distribution"

_materials, 2023, doi:10.3390/ma16134854_

Round 1

Reviewer 1 Report

1. The article title is complex, please refine and better one, you should stress the distortion, due to TIG welding process, and then what type of materials;

2. From Fig. 3 to 17, too many pictures have been provided, could you please collect the most important ones for us, or you can place them together into one figure to reduce the publication space;

3. Microstructrual pictures are bad in quality, please improve them, and then could you please detect the hardness across the joint under varied TIG process.

4. In my opinion, the most important point should be placed on the distortion of thin components, and I think the representation of this aspect is relatively less compared with microstrctures. In addition, you can try the 3D pictures of the distortion.

5. How to define the TIG process, I think it is relevant to heat input? As to this point, could you please try the quantitative analysis in terms of heat input, please refer to the paper "Porosity, element loss and strength model on softening behavior of hybrid laser arc welded Al-Zn-Mg-Cu alloy with synchrotron radiation analysis".

It is good totally.

Author Response

Dear Reviewer
Answers attached

Reviewer 2 Report

This article presents a study about TIG welding distortion on 17-4 PH stainless steel and considering automatic and manual processes. This study is interesting; however, its applications and results are limited since it is considered a single geometric case study and unique process parameters.

The manuscript structure is adequate. Adopted bibliography can be more comprehensive. Some figures required higher resolution and definition, detailed on the specific comments.

It is considered that this article is in the scope of the Materials journal, but it is proposed a major revision before its possible acceptance, aiming to include more information about the study and to improve the text.

Specific comments:

Page 1, Line 2: The designation 17-4PH is not accurate, missing a space: “17 -4 PH”. This is a commercial designation, that should be explained in the article.

Page 1, Line 29: Please describe the application of 17-4 PH stainless steel on aircraft engines. From the information provided, it is not clear the motivation of this study.

Page 2, Line 52: Please verify the variables, that should be in italic. “is the β angle” should be “is the β angle” (please rectify for all variables along the manuscript)

Page 2, Line 55: Please develop the numerical simulation section. Many other case studies can be mentioned and additional references can be considered, including: https://doi.org/10.3390/met10010075 ; https://doi.org/10.1016/j.stmat.2018.08.002 ; https://doi.org/10.1016/j.ctmat.2016.07.015

Page 2, Line 66: “One of the ways to minimize weld distortion is a properly established welding 66 procedure, especially in terms of the solution of the welded structure or the order and 67 method of weld placement.” Please check this sentence, it is not clear.

Page 2, Line 68: “Paper [11] presents the results of numerical simulation of 68 different variants of back-step welds in terms of minimizing weld distortion. The difficulties in achieving adequate dimensional tolerance and shape tolerance” detail the conclusion of this study.

Page 2, Line 89: If the aeronautical sector is the main purpose of this study, identify the engine elements that this steel is used.

Page 2, Line 89: “employed in industrial” should be “employed in industrial machinery “

Page 3, Line 109: “The paper introduces” – please improve the language of the text. “This study introduces”

Page 3, Line 109: Please develop in a deeper ware the objectives of this paper.

Page 3, Line 112: This is the average of how many measurements? Is it according to the chemical composition of the standard?

Page 3, Line 124: “Ampegrage” is “Amperage” ?

Page 3, Line 124: Please describe the procedure to obtain these process parameters. Why only these ones were tested?

Page 5, Line 147: Fig. .2 is warped, please improve it.

Page 6, Line 171: Please adopt the same horizontal scale for all plots, to improve the legibility of the plots.

Page 19, Line 301: “The weld,” should be “The welding lines, “

Page 20, Line 341: “Improve the conclusions/discussion section. They should be more comprehensive.

Page 20, Line 347: “A measurable outcome of the automated welding process is a significant 347 improvement in the quality of the weld face.” This sentence should be rewritten, it is not clear.

The text should be carefully revised to improve the English writing, since that several sentences are not clear. 

Author Response

Dear Reviewer
Answers attached

Reviewer 3 Report

Reviewer comments on the paper

Study of the TIG welding process of thin-walled components made of 17-4PH steel in the aspect of weld distortion distribution by Marek Mróz, BartÅ‚omiej Kucel, Patryk RÄ… and Sylwia Olszewska.

The article presents the results of study on the distribution of weld distortion of thin-walled components made of 17-4PH steel, resulting from TIG welding. Manual and automatic welding processes were examined. Physical simulation of the automated welding process was conducted on a custom-built welding fixture.

Questions:

1.      Line 115. Tab.1 Chemical composition of 17-4PH steel.

It is necessary to indicate in what percentage the composition of the alloy is given (at., wt,).

2.      Is it possible to describe in more detail the method and procedure of “manual welding”.

3.      It seems to the reviewer that the article is overloaded with figures. First of all, this concerns Figures 4-17. Maybe the authors should leave the most important result and describe it to compare manual and automatic welding. A similar remark applies to Figures 20-32.

4.      Line 273. The authors write that areas of different microstructure were also found in the heat-affected zone of the joint, which is the result of the duration of the thermal cycle of welding. What do the authors mean by mentioning the “duration of the thermal cycle of welding”?

5.      Line 322/ Tab. 4. Results of chemical composition analysis of carbides (Figure 23b). It is necessary to indicate in what percentage the EDA data is given (at., wt,).

6.      Line 341. Discussion. The content of the Discussion section is very general. This is not a conclusion or a discussion. The authors need to present more specific conclusions on the work. The conclusion about the advantage of the automatic welding mode would be clear even at the very beginning of the work.

7.      Line 461. Reference 45 is missing from in the text - Tavares, S.S.M.; Da Silva, F.J.; Scandian, C.; Da Silva, G.F.; De Abreu, H.F.G. Microstructure and intergranular corrosion resistance of UNS S17400 (17-4PH) stainless steel. Corrosion Science 2010, 52(11), 3835-3839.

8.      There are no results for “Physical simulation” in this paper. “Physical simulation” is mentioned only in the Introduction (12, 55, 68) and 7 times in the bibliography.

Minor editing of English language required

Author Response

Dear Reviewer
Answers attached

Reviewer 4 Report

This article deals with the distribution of welding distortion against the manual and automatic TIG welding operation in the transverse angular displacement:

1- What is the purpose of including the paragraph at line 102 in the introduction? The treatment is related to the hardness, including relevant references related to welding distortion in different welding processes or other post-treatment work for prevention.

2- The last paragraph of the introduction which highlights the author's novelty and contribution is completely vague. Kindly rewrite it.

3- The work is revolving around the distortion measurement, kindly include real pictures or pictorial illustrations for measurement in the support for Figure 3.

4- How you have obtained the difference graph like Figure 5 and so on? explain it.

5- The authors have just put the figures starting from Fig. 4 to 15, and forget to write the explanation and counter-arguments, why this is happening?

6- What is the difference and significance of Fig 16-17 in contrast with Fig. 4-15?

7- Transerve strain and angular strain are the main working and significance of this article, and it's totally vague, and needs to be modified.

8-How many specimens are prepared for this work, kindly include the real pictures.

9- The authors have put much microstructure and forget to explain the microstructure step by step and everyone. They explained on a random basis, this should be in sequence and in a proper manner.

10- Heading 4 is completely inappropriate, if this is the discussion about the results then must be expended and intensive along with the result and discussion section. If this is the conclusion, must be written again.

Moderate changes are required.

Author Response

Dear Reviewer
Answers attached

Round 2

Reviewer 1 Report

Thanks.

OK.

Author Response

Thanks

Reviewer 2 Report

The authors improved the manuscript accordingly the reviewers' comments.  

Some minor points:

"Graphs of transverse profiles" - "Plots of transverse profiles" 

Use the same type of lines for the plots (Fig. 12 uses a different line compared with Fig. 11 and Fig. 13)

In those Figures, please use a grid, (add vertical lines) and add minor ticks in order to improve the readability. 

Author Response

Dear Reviewer
Answers attached

Reviewer 3 Report

The manuscript is overloaded by figures and illustrations.

No comments

Author Response

Dear Reviewer
Answers attached

Reviewer 4 Report

For every question of round 1, the authors just replied whether they performed the work in some company or it was unnecessary to provide details and explanations regarding the distortion measurements, how they obtained the graphs, a brief description of figures, how specimens were prepared, or an explanation of microstructures; which is unacceptable.

As per the report attached; they have not revised or addressed the queries of round 1. The revision is necessary according to the quality required for the journal. Again a major revision.

Author Response

Dear Reviewer
Answers attached
